# Mechanical Properties and Microstructure of Calcium Sulfate Whisker-Reinforced Cement-Based Composites

**DOI:** 10.3390/ma15030947

**Published:** 2022-01-26

**Authors:** Kai Cao, Ganggui Liu, Hui Li, Zhiyi Huang

**Affiliations:** College of Civil Engineering and Architecture, Zhejiang University, 866 Yuhangtang Road, An-zhong Bldg., Hangzhou 310058, China; lfliugg@zju.edu.cn (G.L.); huili94@zju.edu.cn (H.L.); hzy@zju.edu.cn (Z.H.)

**Keywords:** cement-based composites, calcium sulfate whisker, nanosilica, mechanical properties, microstructure

## Abstract

This study aims to investigate the effect of calcium sulfate whisker (CSW) on the properties and microstructure of cement-based composites. Further, nanosilica (NS) was used as a comparison. The results show that the compressive strength and fracture toughness of cement-based composites increased by 10.3% and 10.2%, respectively, with 2% CSW. The flexural strength, splitting tensile strength, and fracture energy increased by 79.7, 34.8 and 28.7%, respectively, with 1% CSW. With the addition of CSW, shrinkage deformation was aggravated, and the capillary water absorption coefficients were clearly reduced. Compared with NS, CSW-reinforced cement-based composites show better tensile, flexural, and fracture properties and smaller shrinkage deformations and capillary water absorption coefficients. The residual mechanical properties of all groups improve when the treating temperature is lower than 400 °C and decline rapidly when the temperature goes over 600 °C. When treated at 900 °C, the residual mechanical properties are 40% less than those at ambient temperature, with the NS group showing the best performance, followed by the control group and the CSW group. X-ray diffraction (XRD) and scanning electron microscopy (SEM) tests show that the addition of CSW improves the microstructure of the matrix. CSW can reinforce and toughen composites by generating ettringite and whisker pullout, whisker breakage, crack bridging, and crack deflection at the microstructure level.

## 1. Introduction

Cement-based composites are the most widely used building materials in the world [1]. However, there are problems with micro-cracks and micro-defects in cement-based composites after curing due to brittleness—it causes stress concentration and produces a large number of macro-cracks under loading force [2,3]. This characteristic limits wide and practical application [4]. It has been persuasively demonstrated that an effective way to improve the brittleness of cement-based composites is to adopt nano- and micro-materials as additives [5,6]. However, the most widely used nano- and micro-materials in the construction industry, such as nanosilica [7] and carbon nanotubes [8], are expensive. This limits their wide application in practical projects. Calcium sulfate whisker (CSW), the cheapest whisker (about USD 465 per ton), is expected to replace these expensive materials in improving toughness.

CSW, 10 μm to 300 μm in length and 1 μm to 40 μm in diameter, can serve as a kind of micron fiber or ultra-fine inorganic filler. CSW has many advantages, including high strength, excellent toughness, high temperature resistance, and corrosion resistance [3,9]. Therefore, it is widely used in the production of rubber, plastic, paper, construction materials, and in other fields [10]. In addition to being obtained from natural gypsum mines, CSW can also be extracted from industrial by-product gypsum [11]. Hydrothermal [12,13] and atmospheric acidification methods [14,15] are the most commonly used methods of preparing CSW from industrial by-product gypsum. Additionally, studies show that CaSO_4_ whisker can also be synthesized from wastewater using a microwave-assisted [16] or reactive crystallization method [17]. Thus, CSW is a cheap and easily acquired additive.

Essentially, CSW, as a type of low-cost, non-toxic, and environmentally friendly material, shows great potential in a variety of applications, and several studies have reported the application of CSW in cement-based composites. Existing studies have shown that the addition of CSW may increase many mechanical properties of cement-based composites, such as compressive strength [18,19] and flexural strength [20]. In addition, CSW can effectively improve the impermeability, water resistance, and early crack resistance of recycled concrete [21,22]. Such studies show that CSW can defer the setting time of cement when CSW content is below 1.5 kg/m^3^ and shorten the setting time when CSW content is greater than 1.5 kg/m^3^. CSW can increase harmless pores, decrease harmful pores, and optimize the pore size distribution of the matrix [3].

However, there remain questions. First, the effect of CSW on the shrinkage of cement-based composites is unknown, especially when there are no coarse aggregates in the composites [23]. Second, Wan et al. [3] found that there is no chemical interaction between the whisker and cement paste and that the improved mechanical properties are due to the excellent mechanical behavior and pore-filled effect of the whisker. Some other studies have found that CSW can take part in a reaction to generate ettringite during hydration [24]. Thus, it is necessary to find out if there is a chemical interaction between the whisker and cement paste. Third, Cheng et al. [25] investigated the effect of CSW on the high-temperature mechanical properties of calcium aluminate cement-based composites. They found that CSW formed a stable phase and deflected cracks to improve the toughness of the cement binders. However, the effect of CSW on the high-temperature mechanical properties of Portland cement-based composites is still unknown. Lou Chenyang et al. [26] investigated the application of calcium-based whisker in oil-well cement at high temperatures. They found that CSW had no reinforcing and toughening effect on oil-well-cemented rock. This highlights the importance of exploring the high-temperature properties of CSW-reinforced cement-based composites.

To sum up, making full use of CSW is economically and environmentally desirable. However, the application of CSW in cement-based composites has not been extensively investigated, and further research is required. To bridge this gap, the objective of this study is to investigate the mechanical properties and microstructure of CSW-reinforced cement-based composites by adding CSW. Nanosilica (NS), with a large specific surface area and excellent chemical reactivity, is the most widely used nano- and micro-material. In this paper, NS is used as a comparison. The organization of this study is as follows: first, materials and test methods are elaborated. Second, the effects of CSW and NS on the compressive, flexural, and splitting tensile strength of composites are studied. Third, the influence of CSW and NS content on the fracture properties of composites is investigated by a three-point bending test of a notched beam. Additionally, we investigate the shrinkage and capillary water absorption of the composites. Further, high-temperature residual mechanical properties after high-temperature treatment are analyzed. Next, X-ray Diffraction (XRD) tests are used to investigate the phase changes of CSW-reinforced cement-based composites during hydration. The microstructures and enhancement mechanisms of CSW and NS cement-based composites are studied by Scanning Electron Microscopy (SEM).

## 2. Materials and Test Methods

### 2.1. Materials

The raw materials of the cement-based composites used in this study comprised cement, fly ash, CSW, NS, a water-reducing agent, and sand. P•II 52.5R Portland cement (Conch Co. Ltd., Ningguo, China) with a specific surface area of 365.3 m^2^/kg was used. Class-F fly ash with a low calcium content was produced in Jiaxing, Zhejiang Province of China. The physical properties of the CSW (Fengzhu Co. Ltd., Shanghai, China) and NS (Zhitai Co. Ltd., Hangzhou, China) are listed in Table 1 and Table 2, respectively. Polycarboxylate superplasticizer with a water reduction rate of 24–30% was used to improve the fluidity of cement-based composites. The sand used in this study was river sand with a particle size ranging from 0 to 0.6 mm.

### 2.2. Specimen Preparation

To investigate the effect of CSW and NS on mechanical properties, shrinkage, and capillary water absorption of the cement-based composites, Sample C is used as a control group. CSW-1, CSW-2, CSW-3, and CSW-4 are used to denote specimens with a CSW volume percentage of 0.5, 1, 1.5 and 2%, respectively, and NS-1, NS-2, and NS-3 are used to denote specimens with an NS weight percentage of 2, 4 and 6%, respectively. The water/binder ratio used in this study was 0.25, and the sand/binder ratio was 0.42. The amount of water-reducing agent was adjusted according to fluidity.

In this study, the cement-based composites were prepared according to ISO 1920-3-2004 with a cement mortar mixer and the preparation procedure was as follows. First, cement, fly ash, sand, and CSW (or NS) were poured into the mixer and stirred for 90 s. Then, an appropriate amount of water and water-reducing agent were poured into the mixer and stirred for 120 s until the slurry was in a viscous state. Next, the mixture was poured into molds and vibrated for 30 s. After curing for 24 h, the molds were demolded and put into a standard curing room (temperature: 20 °C, relative humidity: 95%) for 28 d.

### 2.3. Test Methods

The compressive strength (size: 70.7 mm × 70.7 mm × 70.7 mm) was tested by an automatic pressure testing machine (YAW-2000, Jinan, China) with a speed of 0.5 MPa/s. The flexural strength (size: 40 mm × 40 mm × 160 mm) and splitting tensile strength (size: 70.7 mm × 70.7 mm × 70.7 mm) were tested by a closed-loop controlled servo hydraulic material test system (WAW-300D, Jinan, China) with a speed of 0.05 mm/min and 0.08 MPa/s, respectively. A shrinkage test was carried out in accordance with JGJ/T70-2009 [27]. The size of the specimens was 40 mm × 40 mm × 160 mm, and there was a copper probe on both sides. After being formed, the specimens were placed in a room at 20 ± 5 °C for 4 h and then smeared. Then, the specimens were moved to a standard curing room (temperature: 20 ± 2 °C, relative humidity: 95%), and the forms were removed after 7 days of curing. Finally, the specimens were moved to another room (temperature: 20 ± 2 °C, relative humidity: 60 ± 5%), and the initial lengths of the specimens were measured after 4 h. The lengths of the specimens were determined as 1, 2, 3, 5, 7, 9, 11, 14, 17, 20, 23, 26 and 28 d.

Three-point bending tests of the notched beams were performed to investigate the fracture properties. The size of the specimens was 40 mm × 40 mm × 160 mm. The fracture toughness *K* and fracture energy *J* can be calculated as below, respectively [28,29].

A capillary water absorption test was conducted based on ASTM C1585 [30], and three specimens with a dimension of 40 mm × 40 mm × 50 mm sawed from the flexural tests were used. First, the specimens were placed in an oven for heating at 60 °C until a constant mass was reached. Second, the four adjacent surfaces of the specimens were sealed with paraffin to bring internal moisture transfer into a one-dimensional path. Before saturating, the original weights of the specimens were recorded by a scale (0.0001 g). Then, the specimens were put into a water tank; the height of the specimens immersed in the water was 5 mm. Finally, the weights at 30, 60, 90, 120, 150, 180, 240, 300, 360, 480, 600, 720 and 1440 min were measured.

Before the high-temperature tests, the specimens were put into an oven (101A, Guangdi Instrument Equipment Co., Ltd., Shanghai, China) at 60 °C and kept for 48 h [31]. Then, a muffle furnace (SX2-4-10A, Suoyu Instrument Equipment Co., Ltd., Shanghai, China) was used to heat the specimens from room temperature to temperatures of 200, 400, 600, 800 and 900 °C, respectively, with the target temperature being maintained for 2 h. Finally, after the specimens gradually cooled to room temperature, the residual mechanical properties were measured.

The X-ray diffractometer (D8 Advance, Bruker, Germany) was used to investigate the effect of CSW on the crystal phase transformation of cement paste. The samples were soaked in absolute ethanol for 24 h and put into an oven at 60 °C for 48 h. Then, the samples were milled until the particle size was no more than 200 mesh. The test range of XRD pattern was from 5° to 90°. The voltage and applied current were 40 kV and 40 mA, respectively.

The field emission scanning electron microscopy was used to observe the changes of microstructure by adding CSW and NS. The samples were placed in an oven at 60 °C for 48 h to stop further hydration. The surfaces of the samples were sprayed in vacuum with gold treatment to improve electrical conductivity. The morphology of the samples was observed by SEM at a voltage of 10 kV.

## 3. Results and Discussion

### 3.1. Basic Mechanical Properties

Figure 1 illustrates the effects of the content of CSW and NS on the compressive strength, flexural strength, and splitting tensile strength of cement-based composites at the age of 28 days.

It can be seen from Figure 1a that compressive strength is slightly increased with the addition of CSW. Compared with the control group, the compressive strength of CSW-4 increases by 10.3%—lower than 15.4% of NS-1. Figure 1b shows that flexural strength is significantly improved with the addition of CSW. Compared with the plain group, the flexural strength of CSW-2 increases by 79.7%—also higher than the NS group. The addition of CSW also improves the splitting tensile strength, shown as Figure 1c. The enhancement of CSW for splitting tensile strength can reach or even exceed NS. The results are consistent with Wan L [3].

With the addition of CSW, some of the CSW forms ettringite, which can fill part of the pores and improve strength. However, excessive dosage will generate too much ettringite, which will lead to excessive volume expansion and cracking. In addition, excessive dosage will also cause uneven dispersion in the matrix and reduce strength.

The dosage of NS has a strong influence on the mechanical properties of cement-based composites. The enhancement of NS for cement-based composites is partly due to the physical filling of NS [32]. Moreover, NS reacts with calcium hydroxide to generate C-S-H gel, accelerating the hydration reaction [33,34]. Then, the structure of the matrix becomes dense, and the mechanical properties are improved. However, when the NS content increases to 6%, the compressive strength and flexural strength are even lower than that of the control group. Excessive NS absorbs a large amount of surface water during mixing, which leads to a reduced degree of hydration. Thus, too much NS forms weak areas in the matrix and reduces its mechanical properties.

The flexural-to-compression ratio can reflect the crack resistance of the composites [35]. As shown in Figure 1d, the flexural-to-compression ratio increases first and then decreases with the increase in CSW and NS content, indicating that the crack resistance of CSW-2 and NS-2 are the highest, respectively.

### 3.2. Fracture Properties

The study of the fracture properties of cement-based composites can not only more accurately estimate the service life and safety of the structure but also provide a basis for measures that can improve the durability of the structure. The fracture toughness and fracture energy of the matrix of the plain group, CSW group, and NS group are shown in Figure 2.

As shown in Figure 2, compared with the control group, the fracture toughness and fracture energy of CSW-reinforced cement-based composites and NS-reinforced cement-based composites increase significantly, while CSW shows a better enhancement effect. NC-3 is even lower than the control group when NS content exceeds 4%. This is consistent with [33].

As a kind of microfiber, CSW can slow down crack propagation by whisker pullout, whisker breakage, crack deflection, and crack bridging [36]. In addition, aciculate and columnar ettringite produced by hydration can play a “bridging” role and form a spatial network structure with the generation of C-S-H gel, which can optimize the microstructure of the matrix and increase its strength. Thus, the toughness of cement-based composites effectively improves, and the expansion of micro-cracks is delayed.

An appropriate amount of NS can effectively play a filling effect as nano-particles and greatly increase the compactness of cement-based composites. Thus, the number and size of primary cracks are reduced. In addition, the possibility of the initiation and propagation of micro-cracks due to stress concentration is also reduced. However, too much NS will produce an agglomeration phenomenon, as it is unable to give full play to the nucleation effect and filling effect of the nanometer particles. A weakness zone is formed inside the matrix that has an adverse effect on the fracture property of the composites [37].

### 3.3. Shrinkage Properties

Due to the evaporation of internal moisture, cement-based composites inevitably produce shrinkage deformation in service. Then, local tensile stress is generated inside the composites, resulting in structural cracking [38]. Since the cement-based composites in this study do not contain coarse aggregate, shrinkage deformation has a more obvious effect on them. Therefore, it is necessary to investigate the shrinkage properties of cement-based composites with a different content of CSW and NS. The shrinkage strain–time curve is shown as Figure 3.

As can be seen from Figure 3, the shrinkage strain of cement-based composites gradually increases with time for the entire control group, CSW group, and NS group. With the addition of CSW and NS, the shrinkage strain is larger than that of the control group. This means that both CSW and NS can increase shrinkage, and CSW-reinforced cement-based composites show lower shrinkage strain than NS-reinforced cement-based composites. The shrinkage strain increases with the increase of CSW content because unreacted CSW absorbs water to generate calcium sulfate dihydrate and calcium sulfate hemihydrate. In addition, water is necessary to generate additional ettringite. It is interesting to notice that the shrinkage strain reaches its maximum when the NS content is 2%. On the one hand, NS absorbs part of the water, making the volume shrinkage larger. On the other hand, NS can accelerate the hydration rate of cement-based composites and fill microscopic pores, resulting in a smaller pore diameter and increased negative capillary pressure [39]. The negative pressure of the capillaries puts the matrix in a state of compression, which causes the appearance volume of the specimen to shrink and the shrinkage deformation to grow. When the content of NS is more than 2%, a large amount of C-S-H gel produced by hydration not only fills the microscopic pores but also turns a large number of open pores into closed independent pores, thus reducing the negative pressure of capillary pores. As a result, the shrinkage strains of NS-2 and NS-3 are slightly lower than that of NS-1.

### 3.4. Capillary Water Absorption Properties

The pore structure of cement-based composites has a very important effect on its mechanical properties [40]. In this study, the capillary water adsorption coefficient obtained through the capillary water absorption test indirectly reflects the internal pores of the composites. The capillary water adsorption coefficient *S* and capillary water adsorption height *i* can be calculated from Equation (1) to Equation (2):
(1)$$i=St^{1/2}+S_{0}$$
(2)$$i=\frac{\Delta m}{A_{0}\rho_{w}}$$
where i is the cumulative capillary water absorption height per unit area of the specimen, $A_{0}$ is the cross section area of the specimen, $S_{0}$ is a constant, Δm is the capillary water absorption mass, $\rho_{w}$ is the density of water, and S is the capillary water adsorption coefficient.

The capillary water absorption properties of the control group, CSW group, and NS group are shown in Figure 4.

It can be seen from Figure 4 that the addition of NS and CSW significantly reduces the capillary water absorption height and the capillary water adsorption coefficient of cement-based composites. The capillary water adsorption coefficient of CSW-2 decreases by 40.3% and that of NS-1 decreases by 36.9%, respectively. The filling effect of CSW and NS has a certain effect on the decrease of porosity. The addition of CSW generates ettringite and fills the pores, which refines the pore distribution and improves the distribution of pores in cement-based composites. In addition, due to the expansion of ettringite, the pores are further reduced. NS reacts with Ca(OH)_2_ to generate C-S-H gel to fill the pores, optimizing the structure of the matrix and making it denser. However, the agglomeration may increase the capillary water adsorption coefficient when there is too much NS.

### 3.5. High-Temperature Residual Mechanical Properties

Fires in tunnels and buildings pose a serious threat to property and safety. Therefore, research on performance degradation and its mechanism in cement-based composites under high temperature is of strong significance for the safe service of cement-based composites and their structures. The residual mechanical properties of the control group, NS group, and CSW group after high-temperature processing are shown in Figure 5.

As can be seen from Figure 5a–c, the residual mechanical properties of the control group, CSW group, and NS group all show a trend of first increasing and then decreasing after high-temperature treatment. When the temperature is lower than 400 °C, the mechanical properties improve due to the promotion of the continuous hydration of unhydrated cement particles and the secondary hydration of fly ash in the high-temperature steam environment. NS generates more C-S-H gel to fill the holes and further improves the performance. When the treating temperature reaches 600 °C, with the decomposition of calcium hydroxide and C-S-H gel, porosity increases, thermal cracking occurs, and the mechanical properties decrease. At this point, the strength is lower than that of the ambient temperature. When the temperature reaches 800 °C, the mechanical properties further decline with the further decomposition of C-S-H and CaCO_3_ [41]. When the temperature continues to rise, cracking and porosity increase further. At the same time, the spherical fly ash begins to melt, and the porous structure inside the fly ash is exposed. After being treated at 900 °C, the residual compressive strength decreases to less than 40% of that of the ambient temperature. In terms of the high-temperature residual mechanical properties at 900 °C, the NS group performs the best, followed by the control group and the CSW group. The decrease of high-temperature mechanical properties is attributed to the decomposition of ettringite.

### 3.6. XRD and SEM Test

XRD was used to investigate the effect of CSW on the crystal transformation of cement paste. Figure 6 presents the XRD patterns of the cement paste with and without CSW.

We can see from Figure 6 that the diffraction peak representing ettringite in the matrix is significantly enhanced after adding CSW, which is consistent with the results of He Y. [24]. It fully illustrates that part of CSW participate in the hydration reaction to generate aft. Then, the pores are filled, and mechanical properties are improved. Other diffraction peaks are basically unchanged.

Changes in microstructure before and after adding CSW and NS were studied using scanning electron microscopy (SEM). SEM images of the control group, NS group, and CSW group are shown in Figure 7.

As shown in Figure 7a, the matrix of the control group is porous, and the bonding effect between interfaces is poor. After adding NS, the microstructure becomes denser, as shown in Figure 7b. This is mainly due to the physical filling effect of NS and its reaction with calcium hydroxide to produce C-S-H gel, filling the internal pores, which makes the structure dense.

Figure 7c shows that many columnar CSWs distribute in three-dimensional disorder in the matrix and are well-bonded to the matrix. The microstructure is denser. On the one hand, micron-sized CSW fills the pores. When the matrix is subjected to the initial tensile stress, there are many microcracks in the matrix. As a type of microfiber, calcium sulfate whisker can bridge cracks and deflect the cracks, thus hindering the expansion of cracks [24]. When the tensile stress in the matrix further increases, whisker pullout and whisker breakage also consume part of the energy that promoted crack propagation, thus slowing down crack propagation.

## 4. Conclusions

In this study, the influence of CSW on the properties and microstructure of cement-based composites was investigated. Based on the experimental results, the following conclusions were reached.

(1) The compressive strength of cement-based composites with 2% CSW increases by 10.3%, while flexural and splitting tensile strength with 1% CSW increase by 79.7% and 34.8%, respectively. The fracture toughness of the samples with 1% CSW increased by 10.2%, and the fracture energy of the samples with 2% CSW increased by 28.7%. Compared with NS, CSW-reinforced cement-based composites show better tensile, flexural, and fracture properties. Too much CSW is harmful to mechanical properties due to excessive volume expansion and cracking of ettringite.

(2) With the addition of CSW and NS, shrinkage deformations are aggravated, and the shrinkage of CSW-reinforced cement-based composites is smaller than that of NS.

(3) With the addition of CSW and NS, capillary water absorption coefficients are clearly reduced. The capillary water absorption coefficients of CSW-reinforced cement-based composites are smaller than those of NS.

(4) The residual compressive, flexural, and splitting tensile strength of all the groups improve when the treating temperature is lower than 400 °C. When the temperature goes over 600 °C, the residual mechanical properties decline rapidly, and the residual mechanical properties are less than 40% of those of normal temperature when the treating temperature is 900 °C. In terms of the residual mechanical properties at 900 °C, the NS group is the best, followed by the control group and the CSW group. The decrease of high-temperature mechanical properties is attributed to the decomposition of ettringite.

(5) XRD tests show a part of CSW in the hydration process generates ettringite that fills pores and increases the strength. SEM images show that the addition of CSW and NS improve the microstructure of the matrix. CSW and NS are well-bonded to the matrix and make the microstructure denser. CSW reinforces and toughens the composites to restrain the formation of cracks through whisker pullout, whisker breakage, crack bridging, and crack deflection on the microstructure level.

## 5. Future Work

The durability (such as anti-chloride ion penetration, anti-carbonization) of CSW-reinforced cement-based composites should be tested in future work to investigate the long-term service performance.

## Figures and Tables

**Figure 1 materials-15-00947-f001:**
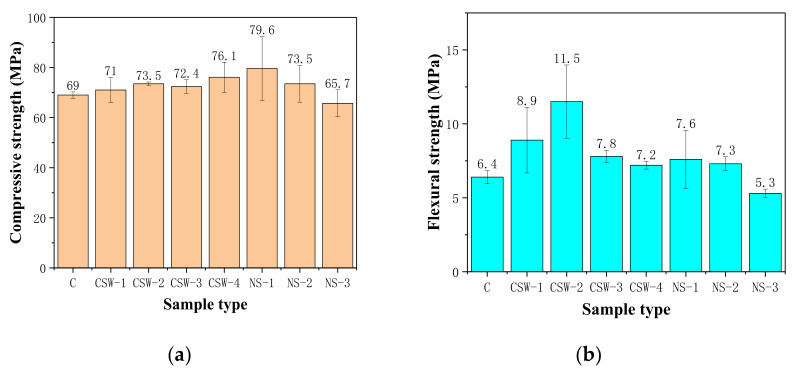
Compressive strength, flexural strength, splitting tensile strength, and flexural-to-compressive strength ratio of the control group, CSW group, and NS group: (**a**) Compressive strength; (**b**) Flexural strength; (**c**) Splitting tensile strength; (**d**) Ratio of flexural-to-compressive strength.

**Figure 2 materials-15-00947-f002:**
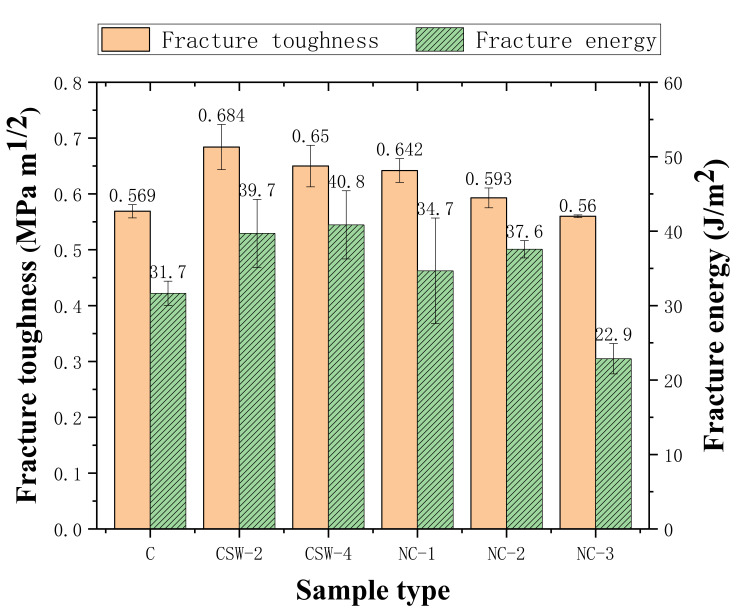
Fracture toughness and fracture energy of the control group, CSW group, and NS group.

**Figure 3 materials-15-00947-f003:**
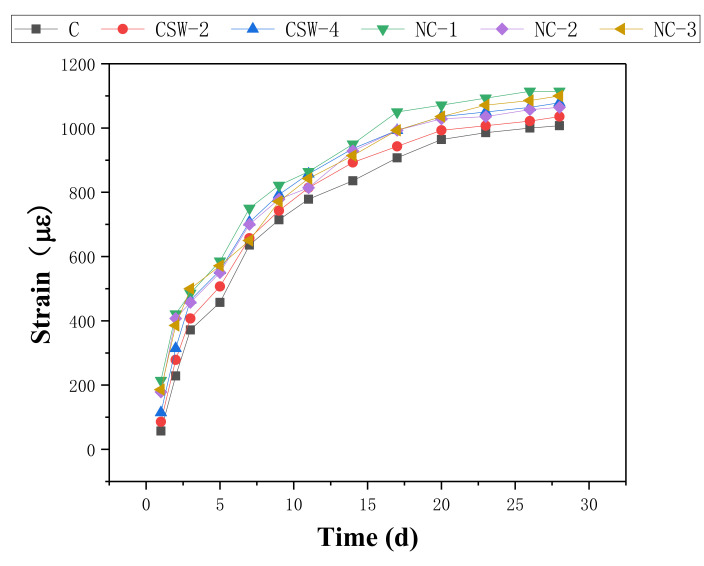
Shrinkage strain–time curves of the control group, CSW group, and NS group.

**Figure 4 materials-15-00947-f004:**
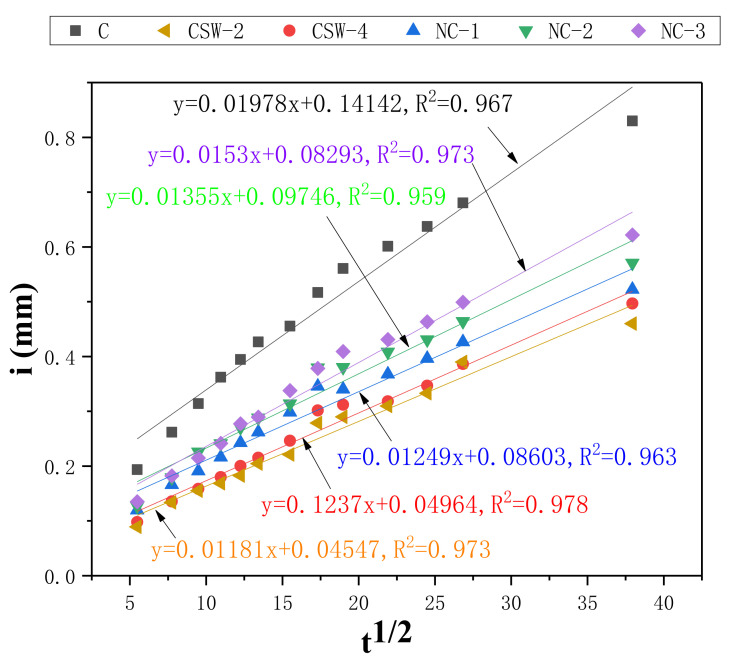
Capillary water absorption properties of the control group, CSW group, and NS group.

**Figure 5 materials-15-00947-f005:**
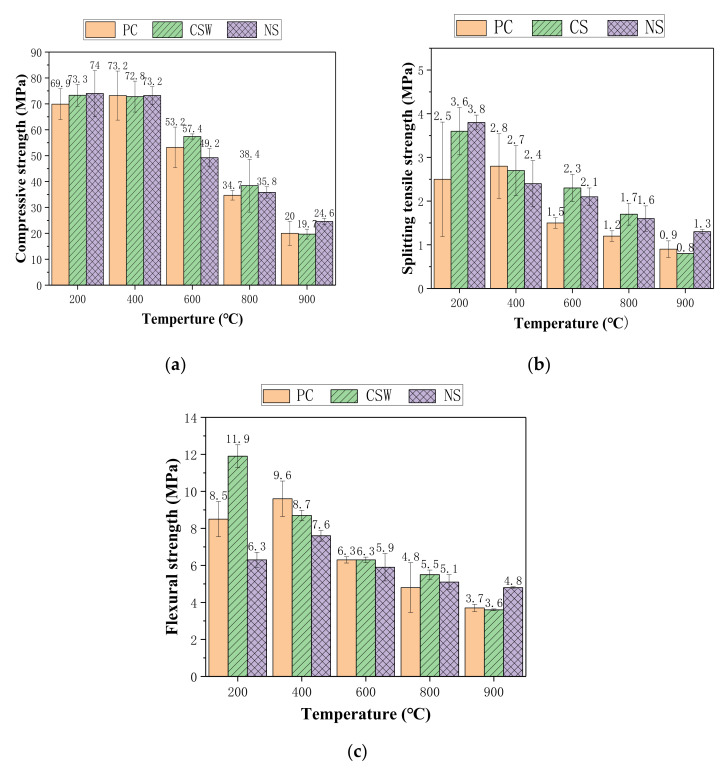
High-temperature residual compressive strength, flexural strength, and splitting tensile strength of the control group, CSW group, and NS group: (**a**) Residual compressive strength; (**b**) Residual splitting tensile strength; (**c**) Residual flexural strength.

**Figure 6 materials-15-00947-f006:**
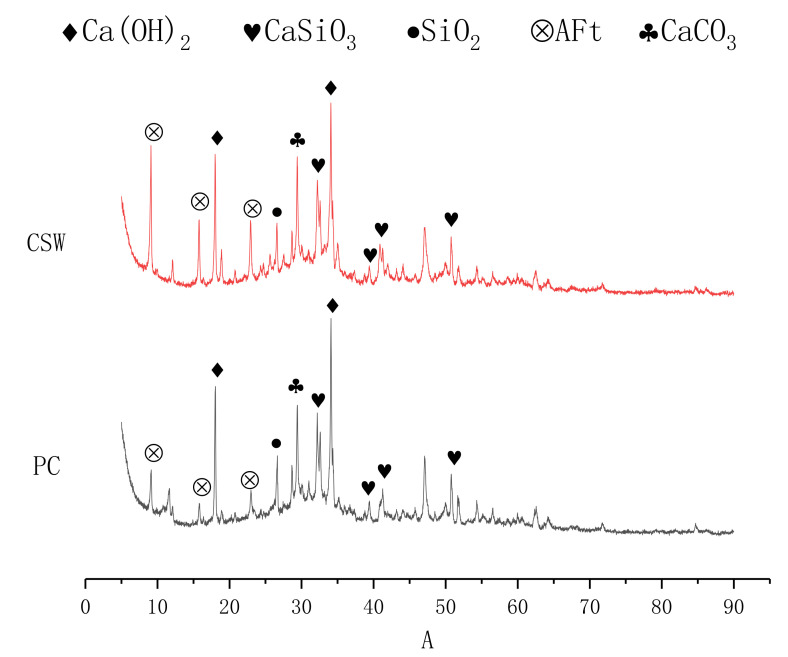
XRD diffractogram of the control group and the CSW group.

**Figure 7 materials-15-00947-f007:**
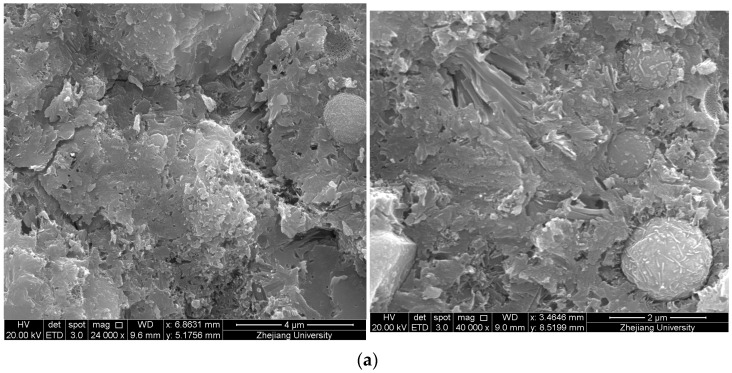
SEM images of the control group, NS group, and CSW group: (**a**) SEM images of the control group; (**b**) SEM images of the NS group; (**c**) SEM images of the CSW group.

**Table 1 materials-15-00947-t001:** Physical properties of calcium sulfate whisker (CSW).

Material	Length (μm)	Diameter (μm)	Moh’s Hardness	Density (g/cm^3^)
CSW	10~50	1~5	3	2.69

**Table 2 materials-15-00947-t002:** Physical properties of nanosilica (NS).

Material	Appearance	Particle Size (nm)	pH	Specific Surface Area (m^2^/g)
NS	White powder	30	5–7	150–300

## Data Availability

The data presented in this study are available on request from the corresponding author.

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
