# Peer review of "Mechanical Properties and Microstructure of Calcium Sulfate Whisker-Reinforced Cement-Based Composites"

_materials, 2022, doi:10.3390/ma15030947_

Round 1

Reviewer 1 Report

Dear Authors,

Overall, this manuscript is interesting. I also believe that it is also quite well done. To improve its quality, you should make a few more important corrections.

Detailed notes below:

Line 8: Each acronym should be clarified before first use.

Line 75: You still have to try to find the scientific basis for your research. That's what's missing here.

Line 76: Considering the above, also refine the purpose of this work.

Line 95: The purchased raw materials as well as the test equipment should be described as follows:
(name of raw material: producer, city, country) or (apparatus model: producer, city, country).

Line 114: Add a mixer model

Line 140: You must add the model of the strength test machine and its selected parameters.

Line 159: Here, too, the test equipment should be described in more detail.

Line 163: The test results are described, but I miss the statistical analysis. The work also does not say what the mustache means (is it a standard deviation, an error ......, then you have to write it). Looking at the error whiskers, I can see that a few samples are statistically insignificant. In general, there should also be a sub-point in the methodology entitled statistical analysis. In principle, you can apply any statistical analysis here, but in my opinion it would be sufficient to use the Tucey or Duncan test. You can review articles where similar methods were used to describe the significance between homogeneous groups and perform the analyzes according to this method:
Properties of Biocomposites Produced with Thermoplastic Starch and Digestate: Physicochemical and Mechanical Characteristics. Properties of Biocomposites from Rapeseed Meal, Fruit Pomace and Microcrystalline Cellulose Made by Press Pressing: Mechanical and Physicochemical Characteristics. Or, Clay / chitosan biocomposite systems as novel green carriers for covalent immobilization of food enzymes. Theoretically, such results can be presented without statistical analysis, but I believe that the quality of the research is then simply lower.

Line 219: Wherever possible, compare your results with those of similar products. This will show if your results are near or far from the intended values. This generally applies to all descriptions of research results.

Line 309: Mark the areas of interest on the SEM images.

Line 349: Add a proposal that will be more forward-looking. A conclusion that will present the perspectives of your research.

Reviewer 2 Report

The manuscript entitled "Mechanical Properties and Microstructure of Calcium Sulfate Whisker-Reinforced Cement-based Composites" presents an interesting experimental study conducted on the obtaining and characterization of cementitious materials with calcium sulfate whisker addition. However, the results of the study have been presented with very limited discussions and other issues must be addressed. The paper needs major revisions before it is processed further, some comments follow:

Introduction section

The introduction section can be improved. The citations have been introduced in bulk form "[1-4]", [5-8], [19-21] etc. and not distributed in the text in accordance with the affirmations that must be supported. Please introduce citation at a specific position to assure a clear correspondence between the affirmations from the introduction section and the previous publication. Moreover, to avoid this type of citing, please cite review type of studies.

Materials and Test Methods Section

Table 2 – particle size 30 nm – is this information correct? All the particles have the same diameter or the particles are lower than 30 nm. Please improve.

Specimen preparation subsection

What was the reason for choosing these parameters? Are these parameters based on previous studies or preliminary experiments? Please introduce corresponding comments.

Please introduce a table to summarize the studied samples and the content of each element from those samples.

Test methods subsection

This subsection is too long and contains multiple unnecessary data/information. As the authors stated, the experiments have been conducted according to standards, therefore, please provide only the standard with a brief description of the samples.

Remove the content from lines 131 to 147 and improve the remaining paragraphs.

The XRD and SEM methods and equipment are missing.

Results and Discussion Section

Figure 3 and Figure 7 – Please provide the graphs in a more compact form.           

XRD – how was the sample preparation for analysis. Have the samples been milled, or only the exposed sample was analyzed?

XRD analysis – There are multiple peaks present in the XRD pattern which haven’t been considered. Why do the authors consider some peaks instead of others? Please improve the description of the XRD spectra.

Mineralogical evaluation. The discussion on XRD patterns is approximative, the identification of some phases is questionable and the evaluation of phase change before and after obtaining the composites wasn’t considered. Please evaluate the samples and provide the experimental results.

Figure 9 – the scientific level of the figures is too low. Please introduce figure labels to highlight the area of interest for the readers.

There is no information about new cause-and-effect relationships identified as a result of the study, in this section. This does not allow forming an opinion about the scientific novelty and theoretical significance of the research results. The experimental results have been presented with very limited discussions.

Please compare the obtained results with those obtained in previous studies and provide a clear overview regarding the novelty introduced by this research.

Conclusion:

The conclusion section can be improved since those are far too general.

Conclusions present some of the results discussed above in the paper with very limited discussion.

Please improve the conclusions and present them following the main recommendations by Academia of giving the conclusions of the study by points with highlights.

References

The reviewed literature is OLD, only four studies published in 2020 (0 from 2021) have been considered (this is way too low). Please conduct a literature survey of the most recent published studies and consider those as the newest discoveries in the field. Please improve the introduction section and the discussions considering these studies.

Reviewer 3 Report

The topic of this work is quite interesting and the whole manuscript is well-prepared, though there are several issues that should be addressed towards the improvement. 

You should provide the explanation of CSW the first time mentioned in abstract as you do in the introduction. In lines 60-61, please provide a reference. In line 81 and 82, you rather use capitals for the words second/third. The last phrase of introduction is unnecessary in my opinion, even if this is not such a significant point. The state-of-the-art is described quite analytically and the whole part of introduction is well-prepared. However, I would recommend to the authors to add a brief general part to your introduction concerning the disadvantages of cement, for which you could find the repsective information among others also in "Chapter 24. Rheological properties of biofibers in cementitious composite matrix" (https://www.researchgate.net/publication/356987531_Advances_in_Bio-Based_Fiber_Moving_Towards_a_Green_Society#fullTextFileContent).

Although the journal you aim to publish this experimental work is international, you have decided to apply non-international standards such as JGJ/T70-2009 and unfortunately, that would be difficult for the readers to follow, repeat or base on your work in future. Which are the differences that this standard have from the ISO ones? Please, provide the details (manufacturer, model) of the oven used (line 150). I could not detect any statistical analysis implemented, neither in materials-methods nor in results section, which is a very significant issue in order to approve the findings and base on them. How do you explain the The manuscript is generally well-organised but the abovementioned issues should be addressed. 

Round 2

Reviewer 1 Report

Dear Authors,

Of course, your article is of great value, so it accepts the corrections made. As for the use of statistical analysis of research results, I believe that they should do so in future research (next articles). These methods are not complicated, and they provide a lot of interesting information regarding the value of your research. Consider this in future articles.

Author Response

Thank you very much for your suggestion.

Reviewer 2 Report

The author adressed most of my comments and the manuscript was improved accordingly. However, figures 1 to 3 have no scientific value as they are presented. 

Figure 1 - does not provide any necessary information to assure experimental repeatability. Also, there is no novelty in this figure. Please remove it and cite the standard according to with was the stirrer manufactured or according to with the mixing stage was conducted.

Figure 2 and Figure 3 - same comments. The experimental setup and the equipment used in the study are ordinary. Please remove the figures.

Author Response

Thank you very much for your suggestion.

1. Figure 1 has been deleted.

ISO 1920-3-2004 Testing of concrete - Part 3: Making and curing test specimens has been added for the mixing stage. (Line 112 in the revised manuscript)

2. Figure 2 and Figure 3 have been deleted.